# Understanding Type 3 Innate Lymphoid Cells and Crosstalk with the Microbiota: A Skin Connection

**DOI:** 10.3390/ijms25042021

**Published:** 2024-02-07

**Authors:** Thao Tam To, Nicole Chizara Oparaugo, Alexander R. Kheshvadjian, Amanda M. Nelson, George W. Agak

**Affiliations:** 1Division of Dermatology, Department of Medicine, University of California (UCLA), Los Angeles, CA 90095, USA; 2Department of Dermatology, Penn State University College of Medicine, Hershey, PA 17033, USA

**Keywords:** ILCs, innate lymphoid cells, ILC3s, skin microbiota, microbiome, skin microbiome, acne vulgaris, *Cutibacterium acnes*, type 3 innate lymphoid cells, skin diseases

## Abstract

Innate lymphoid cells (ILCs) are a diverse population of lymphocytes classified into natural killer (NK) cells, ILC1s, ILC2s, ILC3s, and ILCregs, broadly following the cytokine secretion and transcription factor profiles of classical T cell subsets. Nonetheless, the ILC lineage does not have rearranged antigen-specific receptors and possesses distinct characteristics. ILCs are found in barrier tissues such as the skin, lungs, and intestines, where they play a role between acquired immune cells and myeloid cells. Within the skin, ILCs are activated by the microbiota and, in turn, may influence the microbiome composition and modulate immune function through cytokine secretion or direct cellular interactions. In particular, ILC3s provide epithelial protection against extracellular bacteria. However, the mechanism by which these cells modulate skin health and homeostasis in response to microbiome changes is unclear. To better understand how ILC3s function against microbiota perturbations in the skin, we propose a role for these cells in response to *Cutibacterium acnes*, a predominant commensal bacterium linked to the inflammatory skin condition, acne vulgaris. In this article, we review current evidence describing the role of ILC3s in the skin and suggest functional roles by drawing parallels with ILC3s from other organs. We emphasize the limited understanding and knowledge gaps of ILC3s in the skin and discuss the potential impact of ILC3-microbiota crosstalk in select skin diseases. Exploring the dialogue between the microbiota and ILC3s may lead to novel strategies to ameliorate skin immunity.

## 1. Introduction

Innate lymphoid cells (ILCs) are part of the innate immune system and their major functions include maintaining tissue homeostasis, promoting host defense against pathogens, and regulating tissue remodeling [1]. ILCs provide an early immune response and regulatory function to modify the adaptive and epithelial immune response [2]. ILCs are predominant at mucosal sites and in the intestines, the respiratory tract, lymphoid tissues, and skin. They derive from common lymphoid progenitors (CLPs) and are morphologically similar to T and B lymphocytes [3,4]. The major distinguishing factor of ILCs from adaptive lymphocytes is that they develop in the absence of DNA-rearranging machinery [5]. ILCs also lack rearranged antigen-specific receptors, as observed in T and B cells. However, they express antigen receptors such as Toll-like receptors that are critical in detecting microbial products and inflammatory signals [6]. ILCs can also be categorized based on their expression of distinct transcription factors (TF) and their production of distinct effector cytokines that mirror the classical T helper (Th) cell subsets [7]. The ILC subtypes include natural killer (NK) cells, ILC1s, ILC2s, ILC3s, and ILCregs (Table 1).

ILC1s and NK cells share common characteristics and are grouped as type 1 ILCs [8]. NK cells are cytolytic killers that express interferon-γ (IFN-γ) and target both intracellular pathogens and transformed tumor cells [9,10]. NK cells express high levels of perforin and granzymes and mirror the functions of CD8^+^ T cells, whereas ILC1, ILC2, and ILC3 mirror the CD4^+^ Th1, Th2, and Th17 cells, respectively. Both NK cells and ILC1s express the TF T-box expressed on T cells (T-bet) and secrete IFN-γ and tumor necrosis factor-α (TNF-α). NK cells also share phenotypic markers with both ILC1s and ILC3s, including CD56 and NKp44 in humans and NK1.1 and NKp46 in mice [11,12]. ILC1s possess less cytotoxic activity than NK cells and drive epithelial and matrix remodeling [13]. However, at the same time, IFN-γ and TNF-α production by ILC1s has been associated with microbiota-driven inflammatory responses in inflammatory bowel disease [13,14,15]. 

The microenvironment plays a key role in ILC2 function within peripheral tissues [16,17]. ILC2s express GATA-binding protein 3 (GATA3^+^) and are directly stimulated by epithelial cell-derived cytokines such as IL-25 and IL-33. ILC2s secrete IL-4, IL-5, IL-9, and IL-13, which are critical cytokines in the induction of type 2 inflammation, response to parasite infection, and tissue repair [17,18]. In addition, pollutants, allergens, bacteria, toxins, and dysbiosis of intestinal fungal communities initiate type 2 cytokine responses that enhance ILC2 function and lead to increased severity of allergic diseases such as asthma [19,20].

ILCregs are the most recently discovered subtype, mirroring T regulatory cells (Tregs). ILCregs work to mediate effector functions and dampen inflammation via secretion of IL-10 [21,22]. ILCregs express TFs, Id3, and SOX4, as well as various phenotypic ILC markers such as CD25, CD127, Sca-1, and CD90 [22]. However, ILCregs do not express the signature Treg marker FOXP3 and have not been investigated in the skin [21,22]. 

ILC3s are a heterogeneous category of cells that play a crucial role in intestinal and skin homeostasis, response to infection, and inflammation [23]. Lymphoid tissue inducer (LTi) cells were the earliest discovered ILC3s, and they play a critical role in the formation of lymph nodes and Peyer’s patches [24]. ILC3s uniquely express the surface marker CD117, which differentiates them from other ILC subsets [24]. They also express the TFs RAR-related orphan nuclear receptor-γt (RORγt; RORc) and aryl hydrocarbon receptor (AHR) and secrete IL-17, IL-22, lymphotoxins (LT), and granulocyte macrophage-colony stimulating factor (GM-CSF) [23]. ILC3s can be subdivided into subpopulations based on expression of natural cytotoxicity receptors (NCR) [24,25]. Human NCR^+^ ILC3s express NKp44 and primarily secrete IL-22 and lower amounts of IL-17, whereas NCR^−^ ILC3s secrete IL-17 and lower amounts of IL-22 [24]. 

Overall, ILCs undergo phenotypic changes based on the tissue microenvironment, so the above classification may not completely represent the heterogeneity of this diverse family. ILCs function in response to the microbiota, inflammation, and autoimmunity, and their role in other epithelia-containing organs such as the gut and lungs has been extensively studied. However, the role of ILC3s in skin biology is less clear. ILCs that reside near sebaceous glands (SG) regulate skin microbiome homeostasis and initiate subsequent immune responses [26,27]. ILC3s, in particular, may impact steady-state commensalism by regulating sebocyte function through direct interactions with CD4^+^ T cells [26]. Previous studies have shown that the microbiome composition varies from infancy to adulthood [28]. Our lab has also demonstrated that the skin microbiota is heavily involved in maintaining the skin barrier integrity as well as regulating the complex cellular network within the skin [29]. For example, *Cutibacterium acnes* (*C. acnes*), a predominant commensal in the pilosebaceous unit (PSU), can differentially regulate host Th17 cells [29]. Given the role of ILC3s in modulating epithelial protection against microbes and their close resemblance to Th17 cells, it is plausible that ILC3s may play a role in modulating skin host defense mechanisms and crosstalk with the skin microbiome in inflammatory skin conditions such as acne vulgaris. In this review article, we highlight current evidence linking ILC3s to select skin diseases, with an emphasis on acne. In addition, we discuss how skin-resident immune cells interact with ILC3s to regulate microbial commensalism on the skin and identify current gaps in the knowledge that will guide future research studies.

## 2. Localization and Residency of ILC3s in the Skin

ILC progenitors that differentiate into different ILC subsets in response to specific cytokines have been identified in the peripheral circulation [30]. The total ILC percentage among CD45^+^ cells is lower in peripheral blood than in the skin, but transcriptomic evidence suggests peripheral ILCs can replenish tissue-resident ILC subsets [30]. Both skin and blood ILCs express skin-homing markers such as CCR4 and CCR6, indicating that blood ILCs have the potential to migrate into the skin [31,32]. Despite these similarities, skin ILC subsets are distinguished from their blood counterparts through the expression of tissue residency markers that include CD9, CXCR4, and ITGB4 [32]. In addition, skin-resident ILCs have higher expression of the skin-homing markers CD69 and cutaneous lymphocyte antigen (CLA) in comparison to systemic ILCs [31,32,33,34].

While all ILC populations are found in all layers of healthy human skin, T-bet^+^ ILC1s, RORc^+^ ILC3s, and AHR^+^ ILC3s are the most prominent subgroups [35]. In contrast, higher frequencies of ILC subsets are found in inflamed skin tissues of diseases such as cutaneous atopic dermatitis (AD) and psoriasis [35]. Although ILC numbers are increased in both AD and psoriasis, GATA3^+^ ILC2s and AHR^+^ ILC3s are found mainly in AD lesions, while ILC1s and ILC3s are more visible in psoriatic lesions, highlighting a level of selectivity and specificity of ILCs in inflammatory skin conditions, similar to the classical Th subtypes [35].

Epidermal- and dermal-derived factors contribute to homeostatic maintenance and ILC residency in the skin. As shown in mice, ILCs express IL-7 receptor alpha (IL-7Rα) and require IL-7 receptor signaling via epithelial-derived cytokines, IL-7, and thymic stromal lymphopoietin (TSLP) to remain in the skin [26]. Furthermore, ILCs depend on signaling axes for localization to specific skin niches. For instance, a population of IL-17-producing RORγt^+^ ILC3s is present around hair follicles near the SG [26]. These cells express CC chemokine receptor 6 (CCR6), which binds to its ligand, CCL20, for hair follicle localization [26]. Keratinocytes in the upper hair follicles are the main producers of IL-7 and CCL20 and subsequently promote the recruitment of IL-17-producing RORγt^+^ILC3s [26]. Prior studies demonstrated that ILC3s regulate SG function and commensal bacteria equilibrium, highlighting the importance of keratinocyte-mediated recruitment of ILC3 cells to the PSU [26].

Sebaceous lipid components such as free fatty acids possess antimicrobial activity, hence affecting the viability of microbial species [26]. ILC3s can affect sebum composition in the SG and enhance the commensalism of gram-positive cocci [26]. With *C. acnes* being a prominent gram-positive bacterium in the PSU and using sebum as an energy source, there is a plausible interplay between ILC3s, sebum composition, and *C. acnes*. While ILCs have been described in healthy human skin and are increased in inflammatory skin conditions such as psoriasis, their presence and activity in acne are currently unknown [36,37]. Mechanisms driving the skin-homing and function of ILCs are not yet fully characterized but likely involve skin-derived factors. As such, studies are necessary for addressing knowledge gaps in our understanding of the immune-microbiota crosstalk and how cells interact in the skin to maintain homeostasis but also contribute to disease pathogenesis. 

## 3. ILC3 Interactions with Other Cell Types in the Skin

ILCs interact with immune cells present in both the epidermis and dermis. The cytokine environment shaped by cells in the skin can either regulate immunity or contribute to the pathogenesis of inflammatory skin diseases. Here, we discuss known interactions between ILC3s and other cell types in the skin. We also highlight interactions established in other organs but not yet investigated in the skin (Figure 1). 

### 3.1. Keratinocytes

Keratinocytes can produce molecular factors that are crucial for maintaining epidermal balance and cutaneous inflammatory responses in the skin. Keratinocytes produce cytokines such as IL-7 and CCL20 to recruit IL7R^+^ CCR6^+^ ILCs [26]. IL-7 stabilizes RORγt expression and maintains the ILC3 phenotype to regulate microbial homeostasis [38]. Further, keratinocytes exposed to chronic ultraviolet (UV) light drive expansion of the ILC3-type population and enhance IL-22 and IL-17 production [39]. UV light is a major cutaneous stressor capable of activating hormonal and neuropeptide networks in keratinocytes [40]. These networks trigger homeostatic responses involving effector skin cells, modulate proliferation of keratinocytes, and influence interactions with the local microbiome [40]. While a fundamental interplay exists between keratinocytes, ILC3s, and the skin microbiome, the full extent of their interconnectedness has yet to be established.

### 3.2. Sebocytes

Sebocytes comprise the SG and produce sebum, which contains a mixture of free fatty acids, cholesterols, triglycerides, squalene, wax esters, and antimicrobial substances that protect the skin from external factors, moisturize the surface barrier, and maintain microbial communities [41,42]. Kobayashi et al., revealed that ILCs reside near the SG and influence sebocyte proliferation in the human sebocyte cell line (SEB-1) [26]. Both epidermal and dermal ILC3s negatively regulate sebocyte growth by producing TNF-α and LT, which downregulate factors involved in the Notch signaling pathway that are essential for sebocyte proliferation [26]. In the absence of ILC3s, sebaceous hyperplasia occurs, followed by overproduction of antimicrobial lipids (oleic and palmitoleic acid) that restrict gram-positive bacterial commensalism [26]. Interaction between ILCs and sebocytes is therefore crucial for maintaining balance of the skin microbiota. Given that *C. acnes* are a gram-positive bacterial population that hydrolyze triglycerides in the sebum to release free fatty acids to aid in colonization of microbial populations to the SG, it is worthwhile to investigate the interplay between sebum composition, *C. acnes*, and ILC3s [41,42].

### 3.3. T Cells

ILCs reside in close proximity to T lymphocytes in the skin [35]. Skin-resident ILCs also express IL-16, an alarmin and chemoattractant for CD4^+^ T cells, which suggests a direct interaction between ILCs and T cells in the skin [32,43]. Studies in the gut have shown that ILC3s produce soluble lymphotoxin alpha (LTα3) to promote IgA production and T cell recruitment through TNF receptors [44]. In addition, ILC3s in the gut express high levels of Major Histocompatibility Complex-II (MHC-II) to present antigens and induce commensal bacteria-specific CD4^+^ T cells [45]. To maintain immunologic tolerance and homeostasis of the intestinal microbiota, ILC3s select for RORγt^+^ Tregs and against Th17 cells [46]. The expression of co-stimulatory or co-inhibitory molecules along with MHC-II allows ILC3s to modulate T cell activation in the gut, but whether these interactions and functions occur in the skin are currently unexplored. 

### 3.4. Dendritic Cells (DCs)

DCs also exist near ILC3s in human tonsils, promoting ILC3 proliferation through the production of IL-18 [47]. Proteomic mapping showed evidence of IL-18R1 expression in skin-derived ILCs, suggesting their interaction with DCs via IL-18 [32]. Skin ILC3s express XCL1, a chemoattractant for cross-presenting XCR^+^ DCs (cDC1s), indicating a potential communication between ILC3s and cDC1s [32,48]. In the gut, DCs and ILC3s participate in a feedback loop whereby the production of LT by ILC3s activates LTβR-expressing DCs to produce IL-23 and promote IL-22 responses in ILC3s [49]. In cutaneous leishmaniasis animal models, ILC3-derived IL-17 promotes skin inflammation and this response is directed by CD103^+^ DCs in a *Batf3*-dependent manner [50]. Moreover, this response appears to be regulated by skin bacteria, as skin inflammation is exacerbated in mice colonized with both *Leishmania major* (*L. major*) and *Staphylococcus epidermidis* (*S. epidermidis*), as opposed to infection with *L. major* alone [50]. Additional studies are needed to better understand the connection between DCs, ILC3s, and the interplay of skin microbiota.

### 3.5. Macrophages

Macrophages are key players in wound healing and tissue repair, and it is highly likely that ILCs impact macrophage function (or vice versa) in inflammatory skin conditions. During tissue injury, ILC3s are recruited to the dermis through Notch signaling [51]. As sources of IL-17 and CCL3, these ILC3s promote early macrophage recruitment and epidermal proliferation in wound healing and skin repair [51]. Previous studies have demonstrated that the gut microbiota activates the production of IL-1β by macrophages, which, in turn, stimulates GM-CSF production by ILC3s [52]. This GM-CSF acts on the macrophages to promote differentiation of Tregs and gut immune tolerance. In skin, TREM2 macrophages induced by sebaceous lipids have been shown to contribute to acne [53]. ILC3s may influence this population of macrophages by modulating bacterial communities and sebum production within inflammatory acne lesions; however, these responses have yet to be explored. 

### 3.6. Neutrophils

Neutrophils are active phagocytic cells and are the first cells recruited to the skin at the onset of an infection or tissue damage [54]. For instance, during an *L. major* infection, neutrophils induce IL-17-producing ILC3s to drive early pathology of cutaneous leishmaniasis [50]. Following skin injury, apoptotic neutrophils can initiate tissue repair via the release of lysophosphatidylserine (LysoPS), a direct stimulator of G protein-coupled receptor 34 (GPR34)-expressing ILC3s [55]. In mouse models of airway inflammation, elevated IL-1β levels promote the proliferation of IL-17 and CXCL1-producing ILC3s to recruit neutrophils to the lungs [56]. Elevated levels of IL-1β are linked to a variety of skin pathologies such as psoriasis, acne vulgaris, and cutaneous lupus erythematosus. Thus, it is plausible that ILC3s could function similarly by interacting with neutrophils to drive skin inflammation [57,58,59,60,61]. Overall, more studies are needed to elucidate mechanisms by which neutrophils may initiate, repress, and/or promote crosstalk with ILCs in different skin conditions.

### 3.7. B Cells

ILC3s in the gut, spleen, and mucosal tissues enhance B cell function through CD40 signaling and inducible T cell co-stimulator (ICOS) expression [62]. Splenic ILC3s express B-cell activating factor (BAFF), CD40L, and the Notch ligand Delta Like-1 (DLL1) to activate IgM production in B cells [44,63]. The interaction of ILC3s and B cells at mucosal sites facilitates the further production of cytokines and immunoglobulins, respectively [63]. Although B cells and ILC3s have been shown to co-localize and interact in the gut and mucosal sites, their interactions in the skin have yet to be investigated. 

Ultimately, it is important to study how ILC3s may interact with other cell types to better understand their role in the pathogenesis of skin diseases. Nevertheless, the impact of ILC–immune cell interactions on disease outcomes may be strongly influenced by the local tissue microenvironment. In addition, the balance of the skin microbiota and its interaction with the host affects the skin in both health and disease states. At present, however, little is known about the physiological role and the molecular mechanisms that mediate the interactions between ILC3s and the cutaneous microbiota. Given that acne vulgaris is a disease of the PSU associated with *C. acnes* colonization, and ILC3s are present to regulate sebaceous function and the skin microbiome, we explore below how ILC3 cells may play a role in acne pathogenesis. 

## 4. The Skin Microbiome and Potential Role of ILC3s in Acne Vulgaris

The human skin contains a diverse population of microbial organisms including bacteria, fungi, and viruses which interact with the skin’s innate and adaptive immune cells. These microbiotas, which are body-site-specific and vary among individuals, play an important role in maintaining a robust skin barrier and exhibit relatively stable compositions over time [64]. Importantly, the microbiotas not only promote homeostatic immunity but also provide protection against pathogenic colonization [41]. The skin microbiome is also capable of producing neuromediators such as histamines, glutamate, and various stimulatory neuropeptides, which are instrumental in directing neuroendocrine signaling within the skin [65]. The neurohormonal and neuropeptide networks expressed within skin cells have been reported to help in maintaining epidermal homeostasis, facilitating wound healing, affecting skin aging, and acting in various skin pathologies [40]. In addition, microbial communities residing in the epidermis as well as within the hair follicles and SG modulate the local cytokine environment and contribute to immune responses in a variety of skin disorders [41,42]. 

### 4.1. Skin Bacteria and Acne Vulgaris

The major commensals inhabiting the skin are *S. epidermidis* and *C. acnes* [66]. Commensals protect the skin by preventing colonization of the skin by opportunistic pathogens. Colonization of the skin by pathogenic strains is usually associated with a decrease in the relative abundance of skin commensals [67]. Commensals are critical for maintaining skin health as they secrete antimicrobial agents such as bacteriocins and antimicrobial peptides [68]. In addition, commensals can indirectly inhibit the growth of pathogenic bacterial strains with metabolic products [69]. For example, *C. acnes* can ferment glycerol into free fatty acids, inhibiting the colonization of wounds by pathogenic strains of *S. aureus* [70].

Several studies have demonstrated the ability of skin commensals to modulate the host’s immune response. For example, delta toxin secreted by *S. epidermidis* can induce the formation of neutrophil extracellular traps (NETs) and promote antimicrobial activity against group A *Streptococus* [71]. Furthermore, we recently demonstrated that *C. acnes* strains associated with healthy skin can induce the release of T cell extracellular traps (TETs) from Th17 cells [72]. Remarkably, the same TETs exhibit antimicrobial activity against both gram-positive and gram-negative bacteria, suggesting that both NETs and TETs are involved in the initial antimicrobial responses in acne. However, whether extracellular traps can potentiate ILC activity against invading pathogens remains to be determined. In addition, Nakagawa et al. demonstrated that pathogenic *S. aureus* expresses phenol-soluble modulin, which can induce keratinocytes to secrete IL-1α and IL-36, resulting in IL-17-dependent skin inflammation [73]. This process is intricately connected with the immune response orchestrated by ILC3s, *γδ* T cells, and neutrophils, the primary contributors of IL-17 during *S. aureus* skin infection [73]. Notably, IL-1R and IL-36R signaling via the Myd88 adaptor activates IL-17-producing ILC3s, with these cells serving as key regulators of skin inflammation and exacerbation of skin lesions in response to *S. aureus* [73]. Additional studies are required to elucidate the mechanisms governing the relationship between microbiota-induced extracellular traps, ILC3s, and the skin microbiota.

As major players in barrier tissue homeostasis, ILC3s quickly respond to perturbations through the production of cytokines such as IL-17 and IL-22 [74,75]. Within the skin, IL-22 can modulate the proliferation, migration, and maturation of keratinocytes, and aid in wound healing processes. IL-22 also enhances the antimicrobial responses of keratinocytes through the induction of antimicrobial peptides such as n-defensins 2/3, SA1007/08/09, and lipocalin-2 [76,77]. In line with this, dysregulation of IL-22 and/or IL-22-producing cells is associated with AD and allergic contact dermatitis [77]. We and others reported elevated levels of IL-22 in acne lesions, suggesting that IL-22 can cooperate with IL-17 to activate epithelial cells to produce antimicrobial peptides, but more research is needed to fully understand the role of IL-22 in acne pathogenesis [78,79,80].

Similarly, we reported the presence of IL-17-producing cells in inflammatory acne lesions [78]. Histopathological analysis of skin biopsy samples of inflammatory acne lesions revealed that the frequency of IL-17-producing cells is elevated in acne skin lesions, suggesting that IL-17 is a prominent factor in the disease [78,81,82]. As mentioned previously, studies have shown that IL-17-dependent skin inflammation such as in *S. aureus* infection or in psoriasis involves a range of immune cells, including ILC3s. Moreover, crosstalk between ILC3s and the microbiota is evident and encompasses interactions with the hair follicles and control of sebaceous gland activity [26]. While the role of Th17-derived IL-17 and IL-22 responses have been studied in acne, the role of ILC3-derived IL-17 and IL-22 have yet to be elucidated. Therefore, within the skin microbiome, we envisage a complex and reciprocal interaction between ILCs and the skin microbiota (Figure 2). On the one hand, microbes from acne skin may indirectly influence ILC3 responses through cytokine signals coming from accessory cells such as Langerhans cells and skin epithelial cells, as well as the signaling pathways of microbiota-derived metabolites. On the other hand, ILCs can regulate the skin microbiota by producing effector cytokines such as IFN-γ produced by NK cells and IL-17 and IL-22 produced by ILC3s, as well as the participation of neutrophils and Th17 cells. ILC3s may play a similar role to Th17 cells in enhancing IL-17 production to promote neutrophil recruitment and inflammatory responses or enhancing IL-22 production to promote keratinocyte and fibroblast proliferation within the inflamed acne microenvironment. Moreover, ILC3-derived IL-17 and IL-22 may induce keratinocytes to produce antimicrobial peptides such as cathelicidin and β-defensins that may help maintain skin barrier function [78,82]. CCL20 is upregulated in acne lesions, but whether this chemokine is involved in CCL20-CCR6 signaling by keratinocytes to recruit ILC3s to sites of inflammation in acne remains unknown [82]. 

Furthermore, *C. acnes* can induce skin-resident antigen-presenting cells (APCs), keratinocytes, and sebocytes to produce antimicrobial peptides and proinflammatory cytokines such as IL-1β, IL-6, IL-8, and TGF-β [83,84]. While these cytokines induce naïve T cells to differentiate into effector Th17 cells in acne, IL-6 and IL-1β production may also activate ILC3s and enhance their effector function [29,78]. Although the negative selection of self-reactive T cells in the thymus limits responses to mammalian tissue antigens, the mechanisms that control the selection of *C. acnes*-specific T cells remain poorly understood. Hepworth et al. demonstrated that ILC3-intrinsic expression of MHCII is regulated similarly to thymic epithelial cells and that MHCII^+^ ILC3s may directly induce cell death in activated commensal bacteria-specific T cells [45]. Whether this selection pathway for *C. acnes*-specific CD4^+^T cells occurs in the skin and is dysregulated in acne requires further investigation.

### 4.2. Skin Fungi and Viruses

In addition to bacteria, it is important to consider the consequences of infections from communities of fungi and viruses that may lead to serious dermatological disorders. While skin fungal phylogenetic diversity is low in heathy skin, the most commonly found species *Malassezia* spp. has been shown to convert the skin’s endogenous lipids to free fatty acids and contribute to skin health [28,41,85,86]. Accordingly, DeAngelis et al. demonstrated that *Malassezia* spp. metabolites may contribute to disease severity in skin pathologies including AD and seborrheic dermatitis [86]. Fungal diversity is increased in AD compared to healthy skin, with *Malassezia* aggravating inflammation in AD through IL-23- and IL-17-dependent mechanisms [87]. Subsequently, ILCs are a source of IL-17 in *Malassezia*-exposed skin, promoting inflammatory responses [87]. Another opportunistic fungal pathogen in the skin is *Candida albicans* (*C. albicans*). IL-17- and IL-22-producing ILC3s combat *C. albicans* infection in oropharyngeal and vaginal candidiasis, but this has yet to be explored in skin candidiasis [88,89]. IL-17 plays a key role in fighting *C. albicans* infections by inducing recruitment and activation of neutrophils to clear the fungal pathogens. However, *C. albicans* has been reported to interact with other skin-associated bacterial species such as *S. aureus* to form persistent biofilms, which lead to increased virulence and immune evasion [90,91,92]. Therefore, fungal–bacterial interaction and diversity mediate the host immune response in the skin and may involve IL-17-producing ILC3s. Moving forward, future studies are needed to elucidate the important role ILC3s play in targeting polymicrobial infections. 

While ILC1s mainly respond to viral infection, ILC3s may play a role in defense against viruses [93]. For instance, ILC3s are the source of early IL-17 production in response to adenovirus, and IL-17- and IL-22- producing ILC3s contribute to liver fibrosis in Hepatitis B Virus-related liver cirrhosis patients [94,95]. The oncogenic human papilloma viruses and the Merkel cell polyomavirus (MCV) are the most common viruses found in the skin of adults [96,97,98]. While not much is known about the functions these viral commensals exert on the skin, individuals with compromised immune systems and diminished ILC capacity can be afflicted with tumors associated with MCV [99,100]. Thus, studies providing mechanistic insights into understanding the role of ILC3 response to viral infections in the skin are needed. 

In sum, although ILC3s have been shown to modulate the microbiota and play a role in pathologies that occur in other organs, their role in acne pathogenesis and how they modulate the skin microbiome remains poorly understood. Nevertheless, ILC3s have been investigated in other inflammatory skin conditions and skin infections. Below, we review the role of ILCs in other skin conditions, with the goal of highlighting known roles and mechanisms that may further future research in acne vulgaris.

## 5. ILC3s in Select Skin Conditions

While ILC3s have yet to be investigated in acne vulgaris, these cells have been previously investigated in skin injury, infection, and a variety of skin conditions such as alopecia, psoriasis, AD, hidradenitis suppurativa (HS), and cutaneous leishmaniasis (Figure 3). Thus, we highlight the diverse role of ILC3s in skin immunity in response to infection, injury, autoimmunity, or inflammation.

### 5.1. Alopecia Areata (AA)

The hair follicle encompasses niches for stem cells, immune cells, and microbes. Although the hair follicle is typically an immune-privileged site, conditions such as AA disrupt this environment, leading to a transient, non-scarring form of hair loss that spares the hair follicle [101]. Increased inflammatory and cytotoxic responses driven by T lymphocytes lead to follicular dystrophy and premature entry of the hair follicle into the telogen phase [102,103,104]. While other studies demonstrate that ILCs mediate the hair follicle destruction involved in this disrupted pathway, ILC3-like ILC2s that produce IL-17 are also present in this reaction [104]. Skin-resident ILC3s are located around the hair follicles and persist in this skin niche. These data suggest potential crosstalk between a variety of skin-resident ILCs, T cells, and other immune cells within the inflammatory infiltrates that surround hair follicles and potentially contribute to AA pathogenesis. 

### 5.2. Psoriasis

Psoriasis is a chronic inflammatory skin disease that involves a predominant Th17 response [105]. The quantity of ILC1s and ILC3s, but rarely ILC2s, are expanded in psoriasis [35]. IL-17- and IL-22-producing NCR^+^ ILC3s are increased in both the blood and in human psoriatic lesions [37]. These NCR^+^ ILC3s also express CLA and are increased in non-lesional psoriatic skin compared to normal skin from healthy patients, suggesting their potential role in early disease initiation [78]. Subsequent inflammation caused by Th17 and ILC3 responses in psoriasis can be ameliorated by targeting IL-23 with biologics [106].

Resident ILC3s in psoriatic lesions also express CD200R1, the receptor for the CD200 ligand and an immune regulatory cell surface receptor. Upon ligand engagement, this signaling cascade suppresses myeloid cell activation [107]. IL-17 production is enhanced in ILC3s that are driven by IL-23 and CD200R1 expression [107]. 

Furthermore, psoriasis occurs more frequently in obese individuals, suggesting systemic metabolic involvement in the disease state [108]. In these individuals, monocytes and skin cells have elevated CXCL16 expression, and ILC3s in psoriasis patients have enhanced expression of the CXCL16 receptor, CXCR6 [108]. Proinflammatory ILCs migrate to the skin and promote psoriatic inflammation through CXCL16–CXCR6 interaction [108]. The many roles of ILC3s in psoriasis hence demonstrate their significance in orchestrating skin immunity. 

### 5.3. Atopic Dermatitis (AD)

AD is a very common chronic inflammatory skin disorder often associated with food allergies, asthma, and hay fever. The development of AD involves multiple factors including skin barrier damage and exposure to environmental triggers such as dry heat [109,110,111]. These triggers stimulate the production of epithelial-derived cytokines such as IL-25, IL-33, and TSLP, which promote a predominant Th2 response associated with the release of proinflammatory cytokines such as IL-4 and IL-13 [112]. Further, increased colonization with *S. aureus* and disruption of the skin microbe composition contribute to lesions and disease flares [113,114]. Although ILC2s that exhibit Th2-like responses have been reported in AD skin, ILC3s are the most elevated in atopic skin lesions compared to normal skin [115,116]. The production of IL-1α and IL-23 by inflamed keratinocytes stimulates resident ILC3s to produce the inflammatory cytokine IL-17. Consequently, IL-17 exacerbates inflammation by stimulating IL-33 production from keratinocytes, fostering a type II immune response [117].

### 5.4. Hidradenitis Suppurativa (HS)

HS is a chronic inflammatory condition that causes deep skin abscesses, epithelialized tunnels, and scarring in the skin folds. While the exact pathogenesis of HS is not fully understood, studies have demonstrated that the production of inflammatory cytokines by cells in the epidermis and impairment in Notch signaling are possible mechanisms by which HS develops [118,119,120]. Mutations in the Notch signaling pathway occur in HS lesions and reduced SGs are present in non-lesional HS skin, which could indicate a role for ILC3s in HS pathogenesis through modulation of Notch signaling and sebaceous function [121]. Recent discoveries also reveal that the total number of ILCs is significantly reduced in the blood of HS patients compared to healthy controls [121]. ILCs may thus be migrating from the blood to the skin and contributing to increased inflammation in the skin of HS patients. In addition, the frequencies of ILC3s and ILC2s are increased in non-lesional HS skin, while ILC1s are highest in lesional HS skin [121]. This suggests a role for early initiation of disease by ILC2s and ILC3s and active inflammation in the HS lesion by ILC1s in HS pathology. 

### 5.5. Cutaneous Leishmaniasis

Cutaneous leishmaniasis is characterized by ulcerating skin lesions. IL-17A-producing RORγt^+^ ILCs along with skin microbiota alterations contribute to the pathology of cutaneous leishmaniasis [50]. Inflamed keratinocytes stimulated by *L. major* secrete IL-1α and IL-23, triggering ILC3s to release IL-17, which orchestrates neutrophil recruitment [50,122]. Neutrophils can undergo NETosis as an effective mechanism to combat the parasitic infection. Additionally, dysbiosis in the epidermis in response to the parasite activates ILC1s. ILC1-derived IFN-γ stimulates resident macrophages to release free radicals and reactive oxygen species that directly kill the parasite via oxidative damage [123]. Upon colonization with *Staphylococcus* isolates in *L. major*-infected mice, IL-17-producing RORγt^+^ ILC3s increase in number and immunopathology becomes exacerbated [50]. Such findings provide evidence that the skin microbiota can modulate ILC3 responses to influence disease severity. 

### 5.6. Embryological Immunity

While adaptive immune cells take time to develop, ILCs function to provide critical innate immune protection during embryogenesis and early developmental periods. During neonatal stages in mice, skin-homing CCR10^+^NK1.1^+^ ILC1s control skin commensal bacteria [124]. IL-17-producing ILC3s also regulate immunity during late neonatal and prepubescent stages in mice [125]. The ability of these ILC3s to control cutaneous inflammation and bacterial infection at early stages is dependent on cellular inhibitors of apoptosis proteins 1 and 2 (cIAP1/2) and the canonical NF-κB pathway [125]. Such findings indicate that the skin microbiota can modulate ILC3 responses to influence both disease severity and skin immunity even at early stages of life. 

Although the method by which ILC3s contribute to several skin pathologies is beginning to emerge, their role in other inflammatory skin diseases such as vitiligo, lupus, acne vulgaris, etc., has yet to be explored. The potential contributions of ILC3s in wound healing and hair follicle regeneration also remain an intriguing avenue to be investigated.

## 6. ILC3s in Wound Healing and Hair Follicle Neogenesis

ILCs in the skin express bone morphogenetic protein 1 (BMP1), a metalloprotease involved in extracellular matrix formation and wound repair [32]. Following skin damage, ILC3s in the dermis express homing markers such as CXCR4, CXCR5, and CCR6, allowing them to migrate to the site of injury [51]. Skin injury also activates Notch signaling, which modulates chemokine production in skin cells to recruit immune cells that produce TNF-α and induce the CCL20–CCR6 axis to recruit ILC3s to the dermis surrounding the wound [51]. As mentioned before, these ILC3s contribute to the production of IL-17 and CCL3 to influence macrophage recruitment and initiate the tissue repair process [51] (Figure 3). 

The hair follicle (HF) regeneration cycle is also affected by the wound environment and skin repair process. This complex phenomenon has been extensively studied in murine models, and here, we focus on the role of ILCs in the HF regeneration process (Figure 4). This cycle involves the activity of hair follicle stem cell (HFSC) populations and the intricate coordination of the HFSC niche for SG renewal. As mentioned previously, epithelial-derived signals maintain ILCs that modulate the SG by preventing sebaceous hyperplasia and regulating microbial commensalism [26]. Skin-resident ILCs also express BMP1; BMPs modulate keratinocyte function by inhibiting keratinocyte proliferation in regenerating skin epithelium [126]. Following skin injury, however, perifollicular levels of BMP decline and keratinocyte-derived CCL2 initiate the chemotaxis and recruitment of both CX_3_CR1^+^CCR2^+^ and Ly6C^+^CCR2^+^ macrophages; these macrophages produce several cytokines including TGF-β [126,127,128,129]. Altogether, these cells undergo differentiation into an inflammatory phenotype, leading to the induction of HFSCs through Wnt signaling and TNF-α production [128,129,130]. Activated HFSCs have been reported to aid in the downstream wound healing process by replenishing the SG [131,132].

In typical successive HF cycles, HFSCs in the upper bulge of the HF contribute to sebaceous renewal by differentiating into SG cells in a Wnt/β-catenin-dependent manner [132]. In addition, activation of the Wnt/β-catenin pathway promotes re-epithelialization and the proliferation of keratinocytes and fibroblasts to facilitate wound closure [133]. During the growth phase of the HF cycle, Wnt expression is highest and aids in wound repair, although the opposite effect occurs during the resting phase of the cycle [130]. Following skin injury, however, SG progenitors depart their niche, and all HFSC progenies are recruited to the SG for the replenishment of SG cells, regardless of their location in the HF [131,132]. Furthermore, TNF-α upregulates the activation of Akt/β-catenin signaling and loss of *Pten* expression in HFSCs to promote wound-induced HF neogenesis [130,134]. Inflammation and elevated Akt phosphorylation amplify NF-κB signaling, contributing to the transformation of cells in the epidermis and hair follicle [135]. Simultaneously, while enhanced Akt signaling is associated with promoting tissue regeneration and hair follicle development, it can also lead to epidermal hyperplasia [135].

Given the pivotal role of HFSCs in wound healing and facilitating the regeneration of the HF through renewing the SG, it is important to understand the complexities of this process to prevent hyperplasia. With ILC3s having a key function in modulating sebaceous growth through TNF-α/LT and producing CCL3 to promote early macrophage entry into wounded tissues, these cells may be involved in balancing the wound repair process [26,51]. Further studies are required to elucidate the interplay between HFSCs, SG cells, and immune cells including ILC3s in facilitating the response to skin injury. Ultimately, a balanced wound repair process is essential to minimize infections, prevent excessive scarring, and maintain tissue integrity. In exploring the interplay of ILC3s in different skin conditions, a fascinating dimension appears involving the dynamic realm of cellular plasticity and how these cells can adapt to influence the pathophysiological landscape.

## 7. ILC Plasticity in the Skin

As the largest organ in the body, the skin is in direct contact with the outside environment and requires tightly regulated surveillance mechanisms to keep potentially harmful intruders at bay. For this purpose, the skin is densely populated with ILCs in addition to Langerhans cells, dermal dendritic cell subsets, and macrophages that exhibit dynamic behaviors [136,137,138]. These cells express pattern recognition receptors that make them exquisitely specific in the recognition, uptake, and either direct elimination of pathogenic microbes, or in the presentation of pathogen-associated antigens for subsequent T cell activation [139,140,141]. Even though the phenotype of each subset is clearly defined, environmental cues regulate the interconversion of phenotypes and drive the plasticity of these cells. 

Several extrinsic and intrinsic drivers of ILC plasticity have been defined; however, the cellular metabolic changes that drive plasticity remain largely unknown. As mentioned previously, ILC identity largely depends on their expression of lineage-specific TFs and their cytokine secretion profile. A delicate balance among the TFs determines the frequency of plasticity and hence the fate of ILC subsets. T-bet represses both ILC2 and ILC3 TFs, whereas RORγt antagonizes both NK and ILC1 TFs [142]. The balance in the expression of T-bet, GATA3, and RORγt also determines the fate of NCR^+^ILC3s and NCR^−^ILC3s [143,144]. Taken together, these studies indicate that TFs, through direct control of effector genes or through indirect regulation of other factors, can promote or inhibit ILC plasticity. 

Infection or changing environmental cues and the cytokine milieu leads to the activation of intracellular signaling pathways which influence ILC phenotype and function [105]. For example, the ability of ILCs to undergo plasticity in response to cytokines depends on the surface expression of appropriate cytokine receptors [145,146]. Within the skin microenvironment, activated epithelial cells or myeloid cells produce IL-15, IL-18, and IL-12, which stimulate IFN-γ production through ILC1s, whereas IL-1β, IL-12, and IL-23 can initiate IL-17 and IL-22 production through ILC3s [146,147,148,149]. In mouse models, the conversion of ILC3s to IFN-γ-producing ILC1s can be controlled through the downregulation of RORγt and upregulation of T-bet [38,147,149,150]. Similarly, ILC3s can be converted to ILC1s in humans by IL-1β and IL-12, whereas the combination of cytokines such as IL-1β and IL-23 can reverse ILC1-to-ILC3 conversion [147,151]. Within the gut, conversion of NCR^+^ to NCR^−^ ILC3s depends on both Notch signaling and T-bet [38,144,152,153,154]. Further, the combination of Notch signaling, microbial cues, and IL-23 can lead to the upregulation of T-bet, thereby regulating the development and stability of NCR^+^ ILC3s [144,155]. On the contrary, TGF-β inhibits the conversion of NCR^−^ ILC3s to NCR^+^ ILC3s [146,152]. In addition, TGF-β signaling is involved in NK to ILC1-like cell conversion [156]. However, whether ILC1-like cells can convert and/or transdifferentiate back to NK cells remains unknown. Overall, different cytokine combinations can drive ILC plasticity by inducing distinct phenotypic changes. 

The ability of ILCs to switch between phenotypes highlights their adaptability and apparent role in psoriasis and AD. In vivo fate mapping using fate reporter mice in a psoriasis model has shown an ILC2-to-ILC3 transition and the presence of cells in a mixed ILC2–ILC3 state [157]. ILC3s may arise through different trajectories such as quiescent-like cells or ILC2s [157]. Quiescent-like skin ILCs become ILC3s upon increased accessibility at the TCF7 binding site (ILC precursor cell transcription factor) after induction with IL-23. ILC2-specific genes such as *IL5* and *IL13* are active in cells both before and after induction with IL-23, with ILC3-specific genes such as *IL23R*, *IL22*, and *IL17* showing activity only after induction [157]. Binding site accessibility of ILC3 and ILC2 transcription factors including *RORC*, *BATF*, *STAT3*, and *GATA3* also increases after induction with IL-23 [157]. Bernink et al. demonstrated IL-17-producing ILC3s from psoriatic lesions can convert to ILC2s when cultured with IL-1β and IL-4 [158]. Hence, the development of psoriasis may entail the activation and convergence of tissue-resident ILCs into a pathogenic ILC3 effector state. 

**Table 1 ijms-25-02021-t001:** Table summarizes the ILC subsets in humans, their key cell surface markers, cytokines required for development, cytokines produced, and skin disease prominence. ILC, Innate lymphoid cell; IL, interleukin; CD, Cluster of differentiation; TNF-α, tumor necrosis factor alpha; IFN-γ, interferon gamma; GATA3, GATA-binding protein 3; TSLP, thymic stromal lymphopoietin; AD, atopic dermatitis; GM-CSF, granulocyte-macrophage colony-stimulating factor; NKp44, natural cytotoxicity triggering receptor 2; T-bet, T-box transcription factor 21; Eomes, eomesodermin; NK, natural killer cell; TGF-β, transforming growth factor beta; ID2/3, inhibitor of NDA binding 2/3; SOX4, SRY-box 4; AHR, Aryl Hydrocarbon receptor; CRTH2, Chemoattractant receptor molecule expressed on TH2 cells; ST2, interleukin 1 receptor-like 1; RORγt, RAR-related orphan receptor gamma.

Type of ILC	Key Markers (Human)	Transcription Factors	Cytokines Required for Development	Effector Cytokines	Skin Disease Prominence	References
NK	CD56, CD94, NKp44	T-bet, Eomes	IL-2, IL-12, IL-15, IL-18	IFN-γ, TNF-α, Granzymes, Perforins	Psoriasis, Alopecia Areata, Pemphigus Vulgaris	[75,159,160,161,162,163,164,165,166,167,168,169]
ILC1	CD56 and CD161	T-bet	IL-12, IL-15, IL-18	IFN-γ, TNF-α, Granzyme C	Allergic Contact Dermatitis, Cutaneous Leishmaniasis	[170,171,172,173,174]
ILC2	Lin-Sca-1, ST2, CRTH2, c-Kit^+/−^	GATA3	IL-25, IL-33, TSLP	IL-4, IL-5, IL-9, IL-13, amphiregulin	Atopic Dermatitis, Systemic Sclerosis, Cutaneous Leishmaniasis, Allergic Urticaria	[169,175,176,177,178,179,180,181,182,183]
ILC3/LTi	Nkp44^+/−^ and c-Kit	RORγt and AHR	IL-1β and IL-23	IL-22, IL-17, GM-CSF, lymphotoxins	Acne vulgaris, Psoriasis, Cutaneous Leishmaniasis	[23,52,74,183,184,185,186]
ILCregs	CD25, IL-2Rγ, Sca-1, CD90	Id2, Id3, SOX4	TGF-β and IL-2	IL-10 and TGF-β1	Skin	[21,22]

Similarly, in AD, the plasticity of ILC2s is linked to responses in the skin. Recent single-cell analysis of ILCs from AD skin reveals ILC2 clusters that co-express both type 2 and type 3 ILC markers such as *GATA3*, *RORC*, and *AHR* [31]. Even ILCs isolated from the blood of AD patients co-express ILC2-ILC3 surface markers CD117 and CRTH2 and co-produce ILC2-ILC3 cytokines IL-13, IL-22, and IL-17 upon stimulation with type 2-promoting cytokines [31]. 

Transitioning to another facet of human ILC2-ILC3 behavior, ILC2s isolated from adult peripheral blood, the dermis, and neonatal cord blood can produce IL-17 upon exposure to ILC3 stimuli such as the hyphae of *C. albicans*, or ILC3-promoting cytokines such as IL-1β, IL-23, and TGF-β [158]. This response is associated with the downregulation of the ILC2 markers *GATA3* and CRTH2 and the upregulation of the ILC3 markers *RORC* and CD117. The flexibility of ILCs in changing effector states thus enhances tissue resilience and has physiological relevance in skin pathologies. 

Although ILC plasticity has been demonstrated in a few inflammatory skin conditions, the mechanisms that promote plasticity and the extent of ILC conversion in acne remain largely unknown. Since the conversion of ILC3s and ILC2s into ILC1s requires inflammatory cytokines such as IL-12 and IL-1β, it remains likely that pathogenic strains of *C. acnes* may impact this conversion by changing the cytokine milieu within the PSU in acne. Moreover, the epigenetic circuits that control plastic ILC responses to a changing environment remain largely unknown. It is likely that broad transcriptional and functional changes associated with ILC3-to-ILC1 and ILC2-to-ILC1 conversions are strongly under the control of epigenetic changes, resulting in the activation and silencing of various enhancers. Even though the exact functional impact of ILC plasticity is incompletely understood, it remains likely that plasticity could be one of the mechanisms that enables our immune system to initiate a rapid response to the ever-changing pathogenic skin microbes. Understanding the regulation of TF networks in ILCs could, therefore, lead to the development of new therapeutic targets to treat chronic skin diseases. 

## 8. Concluding Remarks

ILC biology is slightly more than a decade old, and we are still investigating the crosstalk between ILC3s and the skin microbiota. The multifaceted involvement of ILC3s in the skin presents a captivating nexus of immune regulation and tissue homeostasis. Their intricate interplay with the microbiota underscores the relationship between immunity and the skin’s microbial community, shedding light on how all these aspects contribute to skin health. Understanding the functional diversity of ILC3 subsets and their roles in shaping the skin microbiota would be an important avenue to explore. How ILC3 populations respond to microbial signals, mitigate or contribute to dysbiosis, and maintain homeostasis in the skin requires further investigation. 

Targeting the ILC3-mediated pathways holds promise in modulating immune dysregulation and addressing the complexities of skin disease pathogenesis. Looking ahead, the untapped potential of harnessing ILC3 plasticity for precision medicine applications may be possible. By elucidating the molecular mechanisms governing ILC3 function and their dynamic interactions with the skin microbiota, we can anticipate groundbreaking insights that may revolutionize therapeutic approaches, offering new avenues for intervention and promoting skin health.

## Figures and Tables

**Figure 1 ijms-25-02021-f001:**
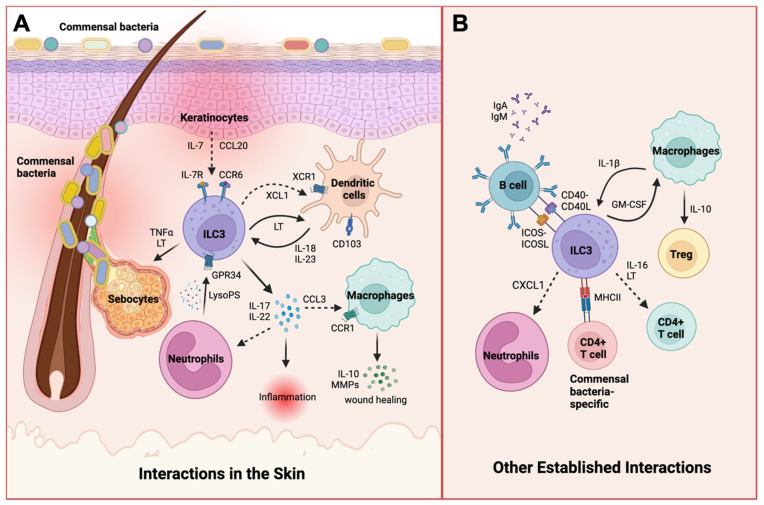
Interactions between ILC3s and other cell types. (**A**) Keratinocytes in the epidermis produce IL-7 and CCL20 to recruit ILC3s. Upon recruitment, ILC3s secrete TNF-α and lymphotoxins (LT) that help regulate sebocyte growth and commensal microbial homeostasis. ILC3s also secrete XCL1, a cytokine important in the recruitment and stimulation of DCs to produce IL-18 and IL-23, activating additional cytokine secretion by ILC3s. Apoptotic neutrophils can also activate GPR34-expressing ILC3s through the release of the metabolite lysophosphatidylserine (LysoPS). Activated ILC3s subsequently secrete IL-17, IL-22, and CCL3 to induce inflammatory responses in the skin or recruit macrophages and promote wound healing. (**B**) Additional interactions that involve microbiota-dependent crosstalk and immune responses with ILC3s have been shown in other epithelial-lined organs such as the gut, lungs, or tonsils, but these interactions have not yet been investigated in skin conditions. For instance, ILC3s can activate immunoglobulin (Ig) production by B cells through CD40-CD40L and ICOS-ICOSL signaling. ILC3s can also recruit neutrophils to inflamed sites through the production of CXCL1. Moreover, macrophages secrete IL-1β during inflammation, stimulating ILC3s to produce GM-CSF and further activating macrophages to induce Tregs that regulate immune homeostasis. ILC3s also reside in close proximity to T cells and can recruit CD4^+^ T cells to target tissues via IL-16 and LT production or express MHC class II to interact with commensal bacteria-specific CD4^+^ T cells. Dashed arrows indicate cellular recruitment. Created with BioRender.com accessed on 13 October 2023.

**Figure 2 ijms-25-02021-f002:**
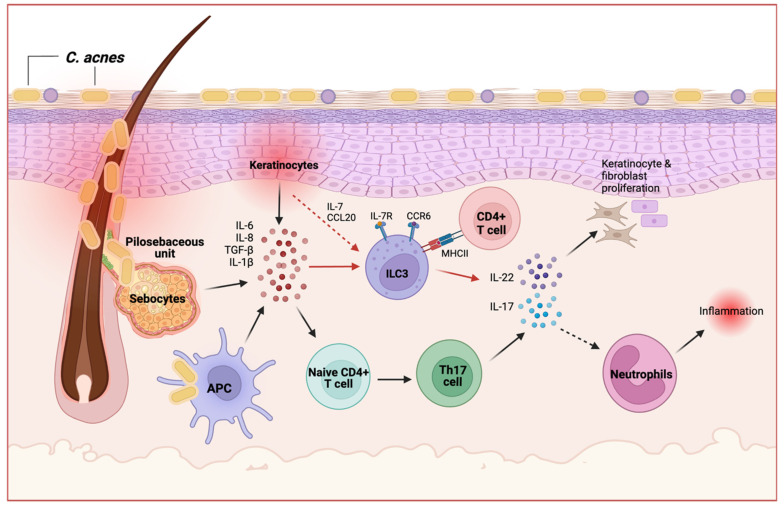
Proposed role for ILC3s in response to *C. acnes*. Commensal *C. acnes* strains activate sebocytes, keratinocytes, and antigen-presenting cells (APCs) to secrete cytokines that induce naïve T cells to differentiate into Th17 cells and initiate a proinflammatory response. Upon *C. acnes* stimulation, ILC3s may also become activated by these cytokines, and keratinocytes may recruit ILC3s through production of IL-7 and CCL20. In close proximity to T cells, ILC3s may activate commensal bacteria-specific CD4^+^ T cells to generate further immune responses against *C. acnes*. Both ILC3s and Th17 cells produce IL-17 and IL-22; these cytokines promote neutrophil recruitment for enhanced antimicrobial response and keratinocyte/fibroblast proliferation, respectively. Red arrows indicate a proposed pathway involving ILC3s in response to *C. acnes*, black arrows indicate pathways established in the literature, and dashed arrows indicate cellular recruitment. Created with BioRender.com accessed on 13 October 2023.

**Figure 3 ijms-25-02021-f003:**
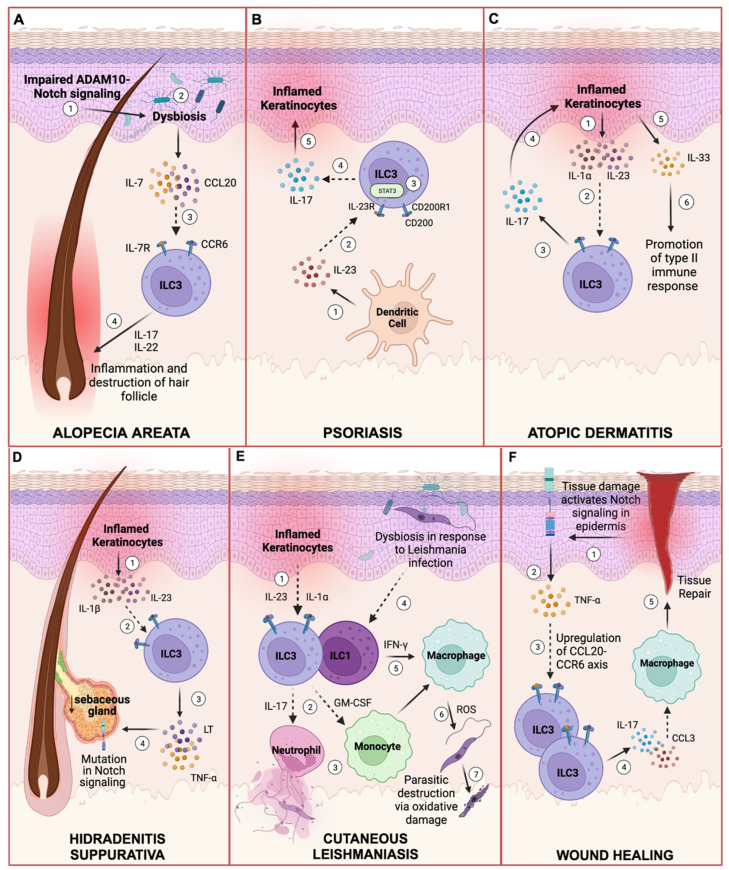
ILC3s have been shown to play a role in several skin diseases. (**A**) Impaired ADAM10–Notch signaling in epidermal keratinocytes leads to dysbiosis of the epithelial microbiota. Surrounding keratinocytes produce chemokines such as IL-7 and CCL20 to recruit and stimulate ILC3s to secrete proinflammatory cytokines that lead to hair follicle destruction in alopecia. (**B**) In psoriasis, ILC3s are activated by IL-23 and the binding of the CD200 ligand to CD200R1. IL-23-driven IL-17 production by ILC3s depends on CD200R1 and signal transducer and activator of transcription 3 (STAT3) activation. (**C**) In atopic dermatitis, inflamed keratinocytes produce IL-1α and IL-23, which recruit and stimulate resident ILC3s to produce IL-17. IL-17 exacerbates inflammation by stimulating keratinocytes to produce IL-33, promoting a type II immune response. (**D**) In hidradenitis suppurativa (HS), inflamed keratinocytes drive inflammation by producing IL-23 and IL-1β. Sebaceous gland reduction in HS skin lesions and disrupted Notch signaling are observed in HS pathology, which may be due to increased ILC numbers in the skin of HS patients or other mechanisms that remain unknown. (**E**) In the parasitic infection of cutaneous leishmaniasis, inflamed keratinocytes also secrete cytokines that stimulate ILC3s to release IL-17. Neutrophils become activated and undergo NETosis to help clear the parasitic infection. Dysbiosis in response to the parasite also prompts ILC1s to produce IFN-ɣ, activating macrophages to eliminate the parasite via oxidative damage. (**F**) Trauma to the skin activates Notch signaling in the epidermis, leading to release of TNF-α and subsequent activation of the CCL20-CCR6 axis and recruitment of ILC3s. Activated ILC3s then release IL-17 and CCL3 to recruit resident macrophages for wound healing and tissue repair. Dashed arrows indicate cell recruitment in the pathways. Created with BioRender.com accessed on 13 October 2023.

**Figure 4 ijms-25-02021-f004:**
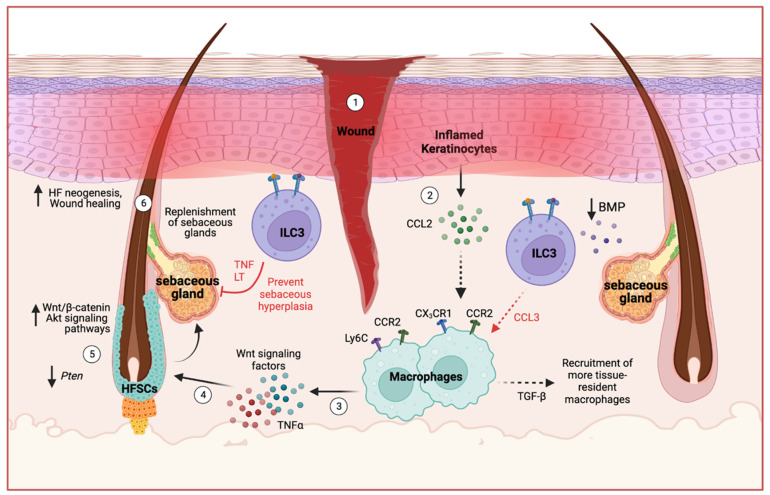
Potential role for ILC3s in wound-induced hair follicle regeneration. Upon injury to the skin of mice, epidermal keratinocytes produce CCL2 to recruit CX_3_CR1^+^CCR2^+^ and Ly6C^+^CCR2^+^ macrophages, with CX_3_CR1^+^CCR2^+^ macrophages producing TGF-β to recruit more tissue-resident macrophages. In addition, skin ILCs express bone morphogenetic proteins (BMPs) that inhibit wound-induced skin repair. BMP levels decrease upon skin injury, however, and epidermal Notch induces recruitment of ILC3s to the wound site where they produce chemokines such as CCL3 that recruit additional macrophages. CX_3_CR1^+^CCR2^+^ and Ly6C^+^CCR2^+^ macrophages undergo differentiation into an inflammatory phenotype, releasing Wnt signaling factors and TNF-α, which activate hair follicle stem cells (HFSCs). Concurrently, the Wnt/Akt/β-catenin signaling pathways become upregulated, while *Pten* expression is lost in HFSCs to promote hair follicle neogenesis and replenishment of the SG. On the other hand, ILC3s produce TNF-α and lymphotoxins (LT) that negatively regulate sebocyte proliferation. Hence, skin ILCs may balance skin wound repair by modulating SG development, maintaining microbiota homeostasis, and coordinating with other immune cells during the wound healing process. Red arrows indicate a proposed role for ILC3s, black arrows indicate pathways established in the literature, and dashed arrows indicate cellular recruitment. Created with BioRender.com accessed on 18 December 2023.

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
