# Peer review of "Understanding Type 3 Innate Lymphoid Cells and Crosstalk with the Microbiota: A Skin Connection"

_ijms, 2024, doi:10.3390/ijms25042021_

Round 1

Reviewer 1 Report

Comments and Suggestions for Authors

I have reviewed the paper by Tam To.

It is erroneous to say that ILCs don’t have antigen receptors, as they have TLRs, which engage PAMPS, and some of these can act as antigens. They lack TCR and BCR, which differentiate them from lymphocytes, if that is what they wanted to say.

Most of the paper is purely speculative trying to link with anything, just by a tangential coincidence with cytokines that are part of general inflammation. For example the Neutrophil passage: “In mouse models of airway inflammation, elevated IL-1b levels promote 262 the proliferation of IL-17 and CXCL1-producing ILC3s to recruit neutrophils to 263 the lungs43. Elevated levels of IL-1b are linked to a variety of skin pathologies such 264 as psoriasis, AD, acne vulgaris, and cutaneous lupus erythematosus, thus, it is 265 plausible that ILC3s could function similarly by interacting with neutrophils to 266 drive skin inflammation.”

There is a lot of forced inclusion f ILC3 based on IL-17, with references that do not back up the claim that this is coming from the ILC3. It can all be a TH17 response. For example the following paragraph has absolutely nothing to to with ILC3. Not even the referenced citation (52).

“Nakagawa et al. showed that pathogenic S. au- 319 reus express phenol-soluble modulin, which can induce keratinocytes to secrete 320 IL-1α and IL-36 to drive IL-17-dependent skin inflammation52. IL-17-producing 321 ILC3s and other immune cells can regulate S. aureus virulence and skin lesion ag- 322 gravation52. Delta toxin secreted by S. epidermidis can induce the formation of neu- 323 trophil extracellular traps and promote antimicrobial activity against group A 324 Streptococus53. In addition, we recently demonstrated that C. acnes strains associ- 325 ated with healthy skin can induce the release of T cell extracellular traps (TETs) 326 from Th17 cells54. Remarkably, the same TETs exhibit antimicrobial activity 327 against both gram-positive and gram-negative bacteria, suggesting that TETs can 328 ameliorate tissue pathology in inflammatory acne. Whether these C. acnes-in- 329 duced TETs can potentiate ILCs activity to increase resistance to invasive skin 330 pathogens such as C. acnes, however, remains to be determined.”

“Similarly, we reported the presence of IL-17-producing cells in inflammatory 345 acne lesions58. Histopathological analysis of skin biopsy samples of inflammatory 346 acne lesions reveal that the frequency of IL-17-producing cells are elevated in le- 347 sional acne skin, suggesting that acne may be an IL-17-driven inflammatory dis- 348 ease58,61,62. In the above studies, however, the role of ILC3-derived IL-17 and IL-22 349 was not determined.”

Table 1 is not appealing

Comments on the Quality of English Language

Article is rather flat.

Reviewer 2 Report

Comments and Suggestions for Authors

The authors begin by providing a classification of ILCs, underscoring their significance in the innate immune system, particularly in preserving tissue homeostasis and responding to pathogens. They meticulously categorize ILCs based on cytokine secretion and transcription factor profiles. The authors highlight the crucial role of skin microbiota in skin host defense mechanisms. They also suggest a potential role of C. acnes in ILC differentiation and plasticity. Subsequently, the authors investigate the potential involvement of ILC3 in acne pathogenesis and other microbiota-related skin conditions.

The authors proceed to elucidate the localization of ILC3 and their interactions with other cell types. They propose that skin microbiota may activate ILC3s, subsequently influencing the microbiome composition, which, in turn, can impact skin health and disease states. Additionally, they stress the need to assess the crosstalk between ILC3s and the microbiota to devise innovative strategies for enhancing skin immunity and treating acne vulgaris.

Further, the authors delve into the role of ILCs, particularly ILC3, in specific skin conditions, and discuss the plasticity of ILCs in the skin. In light of this, the authors express the viewpoint that ILC3-mediated pathways could serve as promising targets for immune dysregulation in skin disease-related pathogenesis.

While the review proves highly resourceful and timely, the exploration of the role of ILCs, especially ILC3, in fungal infections lacks detailed examination. It would be beneficial to include a brief dedicated discussion on this topic, providing general insights.

Reviewer 3 Report

Comments and Suggestions for Authors

The article titled " Understanding Type 3 Innate Lymphoid Cells and Crosstalk with the Microbiota: A Skin Connection" by Thao Tam To and others is a review that primarily explores the impact of wound microbiota on wound healing,

The paper examines the interactions between Type 3 Innate Lymphoid Cells (ILC3s) and skin microbiota, particularly their roles in skin health and disease. The paper covers the critical role of ILC3s in maintaining skin homeostasis, responding to infections, and inflammation, and how they are influenced by the skin microbiome. Results show the involvement of ILC3s in various skin conditions, including their response to microbiota changes and impact on skin health. Paper focuses on the relationship between ILC3s and skin diseases, especially in inflammatory conditions like acne. The conclusion highlights the importance of understanding these interactions and their potential roles in skin disorders. This review is comprehensive, but considering that the International Journal of Molecular Sciences (IJMS) has very high standards for reviews, I believe it has some flaws. I would like suggestions for improvement.

This is a very detailed review, but I still think it needs some additional paragraphs before it can be published.

1.       The article emphasizing and detailedly describes the experimental methods and techniques used for studying the interactions between ILC3s and the microbiota. It discusses the relationship between ILC3s and skin diseases, especially their role in inflammatory skin conditions like acne. This paper the crucial role of interactions between ILC3s and the microbiota, including hemostasis, inflammation, and cell proliferation, and explains how the microbiota influences these processes.

And explores their potential roles in skin diseases. However, I still think it needs additional information on how inflammation affects the sebaceous glands. The image in Figure 3 needs to be supplemented, particularly the part about how TNFalpha affects hair follicle involvement in the sebaceous gland. In Homeostatic Control of Sebaceous Glands by Innate Lymphoid Cells Regulates Commensal Bacteria Equilibrium this paper(Tetsuro,et.al, cell,2019) have clarified ILCs related to TNFalpha, the relationship between hair follicles and wound immunity has been reported in several literatures.CX3CR1 is a Bone derived macrophage, and Ly6C is a Tissue derived macrophage. The relationship between macrophage and -TNF-α and wound needs to be explained, because their downstream pathway is β-catenin(Xusheng Wang, Nature communication,2017) (Waleed Rahmani, JID, 2018). I think it is necessary to cite several literatures to explain the relationship between Wnt/β-catenin in hair follicles and wounds (Xusheng Wang, Nature communication,2017). (Tetsuro Kobayashi Immunity,2019). Regarding the fate between hair follicle stem cells and Wnt/ beta-catenin, I think several articles need to be added with caution(Haiyan Chen,PLOS ONE,2017)(Haiyan Chen,Theranostic,2019).(Xusheng Wang, Nature communication,2017).

WNT/β-catenin pathway promotes the accumulation of HFSCs and the filling of sebaceous glands (SGs). HFSCs and SG cells filled within these glands can further develop into sebaceous cells, playing a positive role in wound healing. Additionally, the hair growth cycle impacts wound repair. During the growth phase, WNT signal expression is at its peak, with HFSCs being active and aiding in wound repair(Xusheng Wang, Nature communication,2017) (Jimin Han iscience., 2023). In contrast, during the resting phase, WNT signal expression decreases, affecting the healing process. About Wnt affects hair follicles to renew sebaceous glands:This is a very new direction, to update the recent literature, we must pay more attention. Hair follicles around Wound would accelerate the renewal of sebaceous glands, and hair follicles could renew sebaceous glands in both normal hair follicles and around wound. Wnt/β-catenin is associated with new sebaceous glands(Jimin Han iscience., 2023)and Sunnywong sequenced sebaceous cells from adult mice. Happily, the article on the relationship between Wnt/ beta-catenin and hair follicles continues(Natalia A. Veniaminova, cell report,2023).

Therefore, when modifying Figure 3 and adding text, it's important to incorporate my suggestions but not to directly copy them.

2.    I think referencing (Sangbum Park, NCB, 2021) in line 560 would be helpful.

Comments on the Quality of English Language

Minor editing of English language required

Round 2

Reviewer 1 Report

Comments and Suggestions for Authors

Authors have responded to my comments with amendments and incorporation of much needed citations.

They acknowledge the still speculative nature of the Review.

Author Response

Authors have responded to my comments with amendments and incorporation of much needed citations.

They acknowledge the still speculative nature of the Review.

Response: Thank you for your valuable comments. We appreciate your time in reviewing our submission.

Reviewer 3 Report

Comments and Suggestions for Authors

It can be published soon.Please cite this paper AKT and its related molecular feature in aged mice skin in here, these cells undergo differentiation into an inflammatory phenotype, leading to the induction of HFSCs through Wnt signaling and TNF- production .

Author Response

It can be published soon.Please cite this paper AKT and its related molecular feature in aged mice skin in here, these cells undergo differentiation into an inflammatory phenotype, leading to the induction of HFSCs through Wnt signaling and TNF- production .

Response: Thank you for your comments. We have incorporated the additional reference as part of the text as shown below.

Edit: "Inflammation and elevated Akt phosphorylation amplify NF-κB signaling, contributing to the transformation of cells in the epidermis and hair follicle135. Simultaneously, while enhanced Akt signaling is associated with promoting tissue regeneration and hair follicle development, it can also lead to epidermal hyperplasia135."

Reference:

(135) Chen, H.; Wang, X.; Han, J.; Fan, Z.; Sadia, S.; Zhang, R.; Guo, Y.; Jiang, Y.; Wu, Y. AKT and its related molecular feature in aged mice skin. PLoS One 2017, 12 (6), e0178969. DOI: 10.1371/journal.pone.0178969  From NLM Medline.

Round 3

Reviewer 1 Report

Comments and Suggestions for Authors

Acceptable

Reviewer 3 Report

Comments and Suggestions for Authors

English language fine. No issues detected. And it can be published.